# Research on Detection Method of Chaotian Pepper in Complex Field Environments Based on YOLOv8

**DOI:** 10.3390/s24175632

**Published:** 2024-08-30

**Authors:** Yichu Duan, Jianing Li, Chi Zou

**Affiliations:** College of Information Science and Engineering, Shanxi Agricultural University, Jinzhong 030801, China; ljnfighting206@163.com (J.L.); 13967180135@163.com (C.Z.)

**Keywords:** Chaotian pepper, intelligent detection, complex environment in the field, YOLOv8, backbone network

## Abstract

The intelligent detection of chili peppers is crucial for achieving automated operations. In complex field environments, challenges such as overlapping plants, branch occlusions, and uneven lighting make detection difficult. This study conducted comparative experiments to select the optimal detection model based on YOLOv8 and further enhanced it. The model was optimized by incorporating BiFPN, LSKNet, and FasterNet modules, followed by the addition of attention and lightweight modules such as EMBC, EMSCP, DAttention, MSBlock, and Faster. Adjustments to CIoU, Inner CIoU, Inner GIoU, and inner_mpdiou loss functions and scaling factors further improved overall performance. After optimization, the YOLOv8 model achieved precision, recall, and mAP scores of 79.0%, 75.3%, and 83.2%, respectively, representing increases of 1.1, 4.3, and 1.6 percentage points over the base model. Additionally, GFLOPs were reduced by 13.6%, the model size decreased to 66.7% of the base model, and the FPS reached 301.4. This resulted in accurate and rapid detection of chili peppers in complex field environments, providing data support and experimental references for the development of intelligent picking equipment.

## 1. Introduction

Pepper is a major vegetable and spice crop globally. Chaotian pepper is a variant of pepper [1] with a small fruit size; generally growing upright or clustering; high pungency; rich in nutrients; and containing capsaicin, carotenoids, phenolic compounds, vitamin C, minerals, etc. [2]. Research has shown that capsaicin substances have preventive effects on various diseases, widely used in medicine, cosmetics, agriculture, and other fields, with promising prospects [3]. Harvesting is a crucial link in the development of the Chaotian pepper industry. Timely harvesting can improve product quality and yield, increase farmers’ income, and reduce labor intensity [4]. The existing harvesting robots are mainly designed for common fruits and vegetables such as oranges, tomatoes, and apples, with regular fruit shapes suitable for mechanical arm harvesting. Currently, a mature harvesting robot for Chaotian pepper has not been developed, and the harvesting is still performed manually. However, traditional manual harvesting is costly in labor, inefficient, and Chaotian pepper plants are located at the top of the plant. Improper manual harvesting can cause damage to the plants, loss of nutrients, etc. Intelligent harvesting of Chaotian pepper can reduce labor costs, increase harvesting efficiency, and promote the intelligent transformation of the industry; thus, the realization of intelligent harvesting is urgent. The detection of Chaotian pepper during the harvesting season is crucial for achieving intelligent harvesting. In the field operation environment, there are issues such as overlapping plants, branches, leaves, fruits, and uneven lighting that affect the accuracy and efficiency of intelligent harvesting [5]. Therefore, researching detection methods for Chaotian pepper in complex field environments are of great significance for achieving intelligent harvesting and promoting the development of Chaotian pepper harvesting robots [6].

Currently, the research and application of deep learning technology in the field of agriculture have achieved fruitful results. Early research mainly focused on traditional methods such as support vector machines [7,8], random forests [5], and HOG-SVM [6,9,10] classifiers. Zhao Jichao et al. [11] designed a harvesting robot system for yellow flower vegetable, which is somewhat similar to Chaotian pepper, using a deep feature binocular camera as the visual system to improve harvesting efficiency. Ma Juanjuan et al. [12] proposed an object detection method based on deep priority random forest classifier, with a detection accuracy of 69%. Zhang Kaibing et al. [13] introduced a method for diagnosing mustard leaf deficiency based on the HSV color histogram of the mustard leaf region, training multiple support vector machine classifiers to achieve leaf deficiency detection, with a mean average precision (mAP) of up to 92.9%. Guo Xin et al. [14] improved HOG and SVM classifiers to achieve multi-target detection of apples, with an mAP of 91%.

Research has shown that changes in natural light intensity [15] and climate [16], alongside issues such as overlapping multiple plants and branches obscuring leaves and fruits [17,18] in complex field environments, have a significant impact on the recognition efficiency of target detection methods. With the widespread application of deep learning [19,20,21,22] technology in the agricultural field, traditional target detection methods with problems of instability and low accuracy have gradually been resolved. Many scholars have conducted detection studies on various crops using models such as Faster R-CNN [23], SSD [24], and YOLO [25,26,27,28]. Guoliang Yang et al. [29] proposed an automatic tomato detection method based on an improved YOLOv8s model using Deep Separable Convolution (DSConv) technology, a Dual-Path Attention Gate (DPAG) module, and a Feature Enhancement Module (FEM), achieving an mAP of 93.4% and a detection speed of 138.8 FPS, thus improving the efficiency of tomato intelligent harvesting. Dexiao Kong et al. [30] proposed an improved YOLOv8s detection algorithm, utilizing an attention mechanism that exploits the sparsity of dynamic query perception by adding a small target detection layer in the neck region and using EIoU as the loss function, and achieved an mAP of 88.5% and accuracy of 94.7% for citrus detection. Zhou Xinzhao et al. [31] established a method for identifying ripe strawberries and detecting fruit stems based on an improved YOLO v8-Pose, adding Slim-neck and CBAM attention mechanism modules, with an accuracy of 98.2%, recall rate of 94.5%, and mAP-kp of 97.91%, which reduced model memory consumption; exhibited good robustness to issues caused by occlusion, lighting, and taking angles; and provided a certain reference value for the intelligent harvesting of bell peppers in field environments. Qiang Luo et al. [32] proposed a strawberry recognition method based on the YOLOv8n model, using the EIOU loss function, using the SPD-Conv method to improve the detection ability of small target point strawberries, with a single fruit detection time of 17.2 ms, which is more suitable for automated strawberry harvesting. Baoling Ma et al. [33] designed a lightweight YOLOv8n model, alternating ShuffleNetv2 and downsampling modules to replace the YOLOv8n backbone module, Ghostmodule for Conv module, and C2fGhost module for C2f module; they achieved a detection accuracy of 91.4%, recall rate of 82.6%, and detection speed of 39.37 fps, providing a reference for the development of fruit and vegetable intelligent harvesting equipment. Shizhong Yang et al. [34] fused YOLOv8s with the LW-Swin transformer module to solve the problem of low recognition accuracy in automatic harvesting and grading of strawberries due to clustering growth resulting in shadows and varying light intensities, achieving a detection accuracy of 94.4% and detection speed of 19.23 fps, and providing a reference for similar issues with cluster-grown bell peppers. Zhou Hongping et al. [35] proposed a camellia fruit classification recognition method based on transfer learning and YOLOv8n for camellia fruit picking robots to address the problem of damage to fruit trees and gripping devices caused by covered camellia fruits by combining learning methods, training data size, learning rate, and number of training rounds; they achieved a highest average precision of 92.7% for YOLOv8n, with a reference value for handling shadow, occlusion, and backlit bell pepper images.

In summary, although crop detection based on deep learning has achieved a series of results, there has been no research report on deep learning in the field of Chaotian pepper detection internationally so far, and the existing detection models have limited application in complex field environments. Based on this, this study proposes a detection method for Chaotian peppers in field environments based on YOLOv8. According to the actual working environment, under different conditions such as normal weather, overcast taken, stem and leaf occlusion with Chaotian peppers, and the mixture of green and red Chaotian peppers, a total of 4038 naturally grown Chaotian pepper samples were collected using web crawling technology. By adding BiFPN, LSKNet, and FasterNet modules to optimize YOLOv8, and continuously optimizing the model by adding different attention and lightweight modules such as EMBC, EMSCP, DAttention, MSBlock, and Faster, and adjusting the model loss function and other parameters, high-precision real-time detection of Chaotian pepper in complex field environments was achieved. This method meets the requirements of real-time and stability in actual working scenarios, aiming to provide an experimental reference and technical support for the intelligent picking of Chaotian pepper.

## 2. Materials and Methods

### 2.1. Data Collection of Chaotian Pepper

#### 2.1.1. Collection Equipment and Methods

Currently, there is no open dataset available for the detection of Chaotian peppers. This study combines data collected through web crawlers and real scene taking to achieve the Chaotian pepper detection task in field environments. A Chaotian pepper dataset covering various conditions was constructed, including normal weather conditions, cloudy weather taking, stems and leaves obstructing Chaotian peppers, and the mixture of green and red Chaotian peppers. The multi-scenario dataset can simulate the actual conditions encountered in the field to the greatest extent and provide a rich set of scenarios for the selection of basic models.

The images in the self-built Chaotian pepper dataset were collected from late July to early August 2023 at the Chaotian pepper planting base in CuSheng, Xiangfen County, Linfen City, China. During the collection process, a handheld Canon EOS M100 mirrorless camera was used to capture the images of Chaotian peppers from top-down, level, and certain angled shots. A total of 1538 images were collected and saved in JPEG format, with a resolution of 1920 × 1280 pixels. The Chaotian pepper data collected using web crawlers were manually selected based on the criteria of image clarity and whether the scene of the image met the requirements of a diverse dataset. A total of 2500 usable Chaotian pepper data images were selected. In this study, the total number of Chaotian pepper images is 4038. The Chaotian pepper images from various scenes are shown in Figure 1a–h.

#### 2.1.2. Data Preprocessing

The input image resolution of the YOLOv8 model used in this study is 1280 × 1280, while the images in the Chaotian pepper dataset have a resolution of 1920 × 1280 pixels. The real-scene collected and crawler-obtained images were resized to 1280 × 1280, and 3000 images were selected from the adjusted dataset as the original dataset. However, due to the complexity of the field environment, having too few data may result in the trained model lacking good stability and generalization ability. Data augmentation can increase the number of datasets, improve the training effect of the model, prevent overfitting, and enhance the generalization ability of the model. Therefore, this study enhanced the original dataset by scaling, rotation, contrast adjustment, brightness adjustment, and mirror flipping, increasing the dataset to 8000 images. The effect of some Chaotian pepper images before and after data augmentation is shown in Figure 2.

#### 2.1.3. Dataset Labeling

This study selected 4000 images for annotation from the data-augmented dataset. The labeling tool used was LabelImg, and the label files were saved in XML format. Based on different scenes of Chaotian pepper in the field environment, the images were annotated with categories of occlusion, sunny weather, cloudy weather, and mixed. Among them, the occlusion category is difficult to distinguish, as occluded Chaotian pepper only shows partial fruit features, which are highly similar to part of the stems, increasing the difficulty of Chaotian pepper recognition.

This study divided the 4000 labeled images into the training set (2800 images), validation set (800 images), and test set (400 images) at a ratio of 7:2:1. The numbers of images corresponding to sunny, occluded, cluttered, and cloudy categories are 2548, 1158, 1136, and 203, respectively. Due to the images being captured in natural environments, each image may contain multiple scenes; so, the total number of images for each category exceeds 4000. The distribution of the 4 labeled scenes in the Chaotian pepper dataset is shown in Table 1.

### 2.2. YOLOv8 Network Model and Evaluation Indicators

#### 2.2.1. YOLOv8 Model Principle and Structure

YOLOv8 is a major update version open-sourced by Ultralytics in January 2023, currently supporting image classification, object detection, and instance segmentation tasks. YOLOv8 is a state-of-the-art model (including target detection networks at P5 640 and P6 1280 resolutions and an instance segmentation model based on YOLACT), built on the successful foundation of previous YOLO versions, introducing new features and improvements to further enhance performance and flexibility. Specific innovations include a new backbone network (the backbone network and Neck section reference the design ideas of YOLOv7 ELAN, replacing YOLOv5’s C3 structure with a more gradient-rich C2f structure and adjusting different channel numbers for different scale models), a new Anchor-Free detection head (the Head section has undergone major changes compared to YOLOv5, adopting the current mainstream decoupled head structure by separating classification and detection heads, and also switching from Anchor-Based to Anchor-Free), and a new loss function (utilizing the TaskAlignedAssigner positive sample allocation strategy for loss calculation as well as introducing Distribution Focal Loss), as shown in Figure 3.

In Figure 3, the UPSample module is an upsampling module that uses the nearest neighbor interpolation method for upsampling. Concat represents tensor concatenation, which expands the dimensions of two tensors. Conv2d stands for two-dimensional convolution. Bbox loss refers to the bounding box loss, which is used to measure the difference between the predicted bounding box and the ground truth bounding box, mainly in object detection tasks. Cls loss represents the classification loss, which is used to measure the discrepancy between the predicted category and the actual category.

#### 2.2.2. YOLOv8 Model Backbone Network

Darknet-53 (53 refers to “52 convolutional layers” + output layer) is the backbone network of YOLOv8, which uses a lot of 3 × 3 convolutions. After each pooling operation, the number of channels is doubled. It uses global average pooling for prediction and places 1 × 1 convolutional kernels between 3 × 3 convolutional kernels to compress features. Batch normalization layers are used to stabilize model training and accelerate convergence. Residual connections are extensively used in the network to alleviate the vanishing gradient problem during training, making the model easier to converge. Additionally, convolutional layers with a stride of 2 are used instead of pooling layers for downsampling.

The backbone network of the YOLOv8 model consists of the CBS module, the C2f module, and the SPPF module. The CBS module is composed of a 2D convolution, 2D BatchNorm, and SiLU activation function, as shown in Figure 4.

The C2f module consists of a CBS convolutional layer and a split, which splits the dimension (c_out) of the feature map of height × width × c_out into two halves, with one half being h × w × 0.5c_out and the other half being h × w × 0.5c_out, as shown in Figure 5.

In Figure 5, Conv refers to the convolution operation, where c1 and c2 represent the input and output of the convolution operation, respectively. k indicates the size of the convolution kernel used. Split represents the split operation, and Concat refers to the channel concatenation operation.

In the convolutional block, c1 is the input channel, c2 is the output channel, and the convolution kernel is 1 × 1; n represents the number of Bottleneck modules. The structure of the Bottleneck module in Figure 5 is shown in Figure 6, which first reduces the dimensionality using a 3 × 3 convolution.

Eliminate redundant information; then, use 3 × 3 convolution to upsample. In the first convolution layer f, c1 is the input channel, c_ is the output channel, and the corresponding convolution kernel is 3 × 3 with a stride of 1; in the second convolution layer f, c_ is the input channel, c2 is the output channel, and the corresponding convolution kernel is 3 × 3 with a stride of 1; when c1 and c2 are equal, residual connection is used.

The SPPF structure, as shown in Figure 7, consists of a CBS convolutional layer followed by three sequential Maxpooling operations. The feature maps before Maxpooling and those generated after each additional Maxpooling operation are concatenated to achieve feature fusion.

In Figure 7, MaxPool2d represents the two-dimensional max pooling operation, where k=(5×5) indicates that the kernel size used for the max pooling operation is 5 × 5. Concat refers to the channel concatenation operation.

In the convolutional block f, c1 represents the input channel, c_ represents the output channel, and k represents the number of convolution kernels. The convolution kernels in the convolutional block are of size 1 × 1, while the convolution kernels in the Maxpool have a size of 5 × 5.

#### 2.2.3. Other Parts

The Neck part uses the C2f module as a residual block. The PAN-FPN first downsamples and then upsamples, with two cross-layer fusion connections between the upsampling and downsampling branches. The Head Structure of the Network first forks into two CBS convolutional modules, then connects a Conv2d, and finally calculates the classification loss and Bbox loss separately. The loss function computation consists of two branches: classification and regression branches, where the classification branch uses BCE Loss, as shown in Formula (1).
(1)$$Loss=-\frac{1}{N}\sum_{i=1}^{N}y_{i}*log(p(y_{i}))+(1-y_{i})*log(1-p(y_{i}))$$
where y is a binary label of 0 or 1, p(y) is the probability of the output belonging to the label, and N represents the number of groups of objects predicted by the model. The regression branch uses Distribution Focal Loss, which is calculated as shown in Formula (2).
(2)$$DFL(S_{i},S_{i+1})=-((y_{i+1}-y)log(S_{i})+(y-y_{i})log(S_{i+1}))$$

The model also uses CIoU Loss, which is calculated as shown in Formula (3). The three losses can be weighted using specific weight ratios.
(3)$$CIoU=IoU-\frac{D_{2}^{2}}{D_{C}^{2}}-\alpha v$$

The weight hyperparameter is calculated according to Formula (4).
(4)$$Loss=-\frac{1}{N}\sum_{i=1}^{N}y_{i}*log(p(y_{i}))+(1-y_{i})*log(1-p(y_{i}))$$

According to Formula (5), h and w represent the length and width of the rectangle.
(5)$$v=\frac{4}{\pi^{2}}{\left(arctan\frac{w^{gt}}{h^{gt}}-arctan\frac{w}{h}\right)}^{2}$$

### 2.3. Design of the Main Optimization Modules

In this study, the data of Chaotian pepper in the field environment were used, captured from multiple angles. The size of Chaotian pepper fruits varied. Based on the characteristics of this dataset and the practical application requirements in production, this study employed modules that can integrate the multi-scale information of images, attention mechanisms, and modules suitable for lightweight applications. The effects of different optimization designs on the basic model were compared and integrated. Some of the key optimization modules are introduced as follows.

#### 2.3.1. BiFPN Optimization Module

BiFPN (Bidirectional Feature Pyramid Network) is the abbreviation for Bidirectional Feature Pyramid Network, which is a neural network architecture used in computer vision for object detection and segmentation tasks. BiFPN introduces bidirectional flow of feature information, effectively solving the problem of information loss in traditional feature pyramid networks when extracting features of different scales. BiFPN introduces learnable weights and, through layer-wise fusion of upsampled and downsampled feature maps, introduces lateral and vertical connections, allowing features of different scales to be better integrated and utilized.

Traditional FPN network structure merges features by weighting feature maps of different scales equally. However, for feature maps of different resolutions, using the same weight leads to uneven feature maps in the output. Therefore, BiFPN sets different weights based on the priority of different input features and iteratively applies this structure to enhance feature fusion. It mainly introduces a bottom-up and top-down feature propagation mechanism, allowing information to flow bidirectionally between different levels, facilitating the comprehensive integration of high-level and low-level features, thereby achieving weighted fusion and bidirectional cross-scale connections. The structure of BiFPN is shown in Figure 8.

From Figure 8, it can be seen that the feature maps of p3, p4, p5, p6, and p7 are adjusted in channel through 1 × 1 convolution to obtain the feature inputs P3_in, P4_in, P5_in, P6_in, and P7_in. The intermediate features P6_td, P5_td, P4_td, and P3_out are the output feature of the third level. The green curved arrows represent P4_in2, P5_in2, and P6_in2, which indicate the cross-scale connections from the inputs of P4, P5, and P6, respectively, directly to the outputs of each level. The third-level output feature P3_out, which has been upsampled layer by layer, is downsampled and stacked with the cross-scale connection P4_in2 and the intermediate feature P4_td to obtain the fourth-level output feature P4_out. The fifth-level output feature P5_out is obtained by downsampling the fourth-level output feature P4_out and stacking it with the cross-scale connection P4_in2 and the intermediate feature P5_td. The sixth-level output feature P6_out is obtained by downsampling the fifth-level output feature P5_out and stacking it with the cross-scale connection P6_in2 and the intermediate feature P6_td. The seventh-level output feature P7_out is obtained by downsampling the sixth-level output feature P6_out and stacking it with the feature input P7_in.

The blue arrows P7_U, P6_U, P5_U, and P4_U represent the top-down upsampling process, conveying semantic information of high-level features. The orange arrows P3_D, P4_D, P5_D, P6_D, and P7_D represent the bottom-up downsampling process, conveying positional information of low-level features, making them robust in dealing with scale changes and occlusions in complex scenes.

BiFPN integrates feature maps from different layers efficiently through a normalization fusion method, enhancing the model’s computational efficiency. Specifically, BiFPN utilizes a weight-based normalization method, where each weight $w_{i}$ is divided by the sum of all weights $\sum_{j}w_{j}$ to ensure that these weights are normalized within the [0, 1] range, thereby accelerating computation. This normalization process is expressed in Equation (6) as follows:(6)$$O=\sum_{i}\frac{w_{i}}{\in +\sum_{j}w_{j}}\cdot I_{i}$$
where O denotes the output feature map,$w_{i}$ and $w_{j}$ are weights obtained during the network training process, and $I_{i}$ represents the input feature map. To ensure the non-negativity of the weights, the ReLU activation function is used. Since scalar weights might cause training instability, BiFPN employs the softmax function for weight normalization. The feature fusion process in BiFPN is described in Equations (7) and (8) as follows:(7)$$P_{i}^{m}=Conv\left(\frac{w_{1}\cdot p_{i}^{in}+w_{2}\cdot Res\left(p_{i+1}^{in}\right)}{w_{1}+w_{2}+\in }\right)$$
(8)$$p_{i}^{out}=Conv\left(\frac{w_{1}^{\prime }\cdot p_{i}^{in}+w_{2}^{\prime }\cdot p_{i}^{m}+w_{3}^{\prime }\cdot Res\left(p_{i-1}^{out}\right)}{w_{1}^{\prime }+w_{2}^{\prime }+w_{3}^{\prime }+\in }\right)$$

Here, $P_{i}^{m}$ represents the intermediate feature layer from top to bottom and $p_{i}^{out}$ denotes the output feature layer from bottom to top. Res stands for the upsampling or downsampling operation, and Conv is the convolution operation. Through this bidirectional feature fusion operation, BiFPN can more efficiently integrate multi-scale features, thereby improving detection speed and accuracy. These mechanisms enable BiFPN to enhance the flexibility of feature fusion and significantly improve the model’s performance.

#### 2.3.2. EMSCP Optimization Module

EMSCP (Efficient Multi-Scale-Conv plus), compared with conventional convolution, not only reduces the number of parameters and computations but also improves detection accuracy, achieving lightweight and efficient object detection. EMSCP is a compact convolution module that integrates multi-scale information. Its main design inspiration comes from the concept of group convolution [36,37]. EMSCP is a multi-scale convolution network architecture that captures different features in the image by applying multi-scale convolution operations on inputs at different scales. This architecture concatenates feature maps at different scales to obtain a richer feature representation. It helps to enhance the model’s perceptual and expressive capabilities, and it is especially suitable for processing images with features at different scales. The EMSCP structure is shown in Figure 9, starting from applying ordinary convolution operations to the feature maps. These maps are then divided into m groups, and linear operations Φ are performed on each group to generate complete feature maps (φ is set to 1 × 1, 3 × 3, 5 × 5, or 7 × 7 convolution kernels). Subsequently, channel-wise feature fusion is performed using pointwise convolution, resulting in the output.

Standard Convolution refers to the regular convolution operation used to extract both local and global features from pooled image data. The variable m represents the grouping of feature maps, and φ denotes the linear operation performed on the grouped feature maps, typically using convolution kernels of sizes 1 × 1, 3 × 3, 5 × 5, and 7 × 7. Pointwise Convolution, on the other hand, refers to the 1 × 1 convolution, which is used in multi-channel convolutions to weight and combine feature maps along the depth direction, generating fused feature maps.

This model replaces the 3 × 3 convolution in Bottleneck with EMSCP as the main gradient flow branch using class inheritance, reducing model floating-point numbers and computational complexity. The model is named Bottleneck-EMSCP. The model structure is shown in Figure 10.

In Figure 10, h represents the height of the feature map, w represents the width of the feature map, and c represents the number of channels in the feature map. When ‘add: true’, it indicates that a residual connection (or skip connection) is included in the module. When ‘add: false’, it means that no residual connection is used in the module.

By constructing C2f-EMSCP, EMSCP is introduced to further reduce the floating-point numbers and computational complexity during the convolution process. The structure of the C2f-EMSCP module is shown in Figure 11.

In Figure 11, h represents the height of the feature map, w represents the width of the feature map, and c represents the number of channels in the feature map, which can refer to color channels or the depth of the feature map. n represents the batch size. Convmodule refers to modules or blocks in the network that perform convolution operations. Split indicates the splitting of a feature map into multiple parts. Concat represents the concatenation along the channel dimensions.

The optimized Bottleneck-EMSCP and C2f-EMSCP modules are fused into the backbone network of the basic model to form the optimized overall structure. Through the addition and optimization of different modules, the goal is to achieve a comprehensive improvement in model efficiency. Its structure is shown in Figure 12.

### 2.4. Experimental Environment Setup

In this study, the research group conducted experiments on a device running Windows 10 Enterprise LTSC operating system. The device is equipped with an Intel^®^ Core™ i7-13700F 2.1 GHz processor and 32 GB of RAM. Additionally, the system is equipped with an NVIDIA GeForce RTX 4080 graphics card. The experiment used CUDA version 11.8, PyTorch 2.0.1 for deep learning framework, and Python version 3.9.

### 2.5. Model Parameter Settings and Evaluation Indicators

The experiment used YOLOv3, YOLOv5, YOLOv6, and YOLOv8 as object detection models; optimized the network model using the Adam optimizer; set the input image resolution of the model to 1920 × 1280 pixels; and set the iteration number for model training to 200, batch size for each iteration to 8, and the initial learning rate to 0.001. The specific model training hyperparameters are shown in Table 2.

The performance of object detection models is usually evaluated using metrics such as precision, recall, mean average precision (mAP), and frames per second (FPS). Precision can be used to evaluate the model’s ability to recognize detected objects, while recall can be used to assess the model’s ability to retrieve true positive samples. By recording precision and recall values, a PR curve can be plotted. The average precision (AP) of a model in detecting objects is the area under the PR curve, which can be used to evaluate the overall performance of the model in object detection and classification. The mean average precision represents the average of all class-specific average precisions, providing a more accurate reflection of the overall performance of the model in detecting various classes of objects. Frames per second refers to the number of frames a network model can process per second, measuring the speed at which the model processes images. Therefore, the evaluation of object detection models in this experiment mainly focuses on the following metrics: precision, recall, mAP, and FPS. The formulas for calculating precision, recall, AP, and mAP are shown in Equations (9)–(12).
(9)$$Precision=\frac{TP}{TP+FP}$$
(10)$$Recall=\frac{TP}{TP+FN}$$
(11)$$AP=\int_{0}^{1}P(R)dr$$
(12)$$mAP=\frac{1}{n}\sum_{i=1}^{n}AP_{i}$$

Among them, TP represents the number of true positive instances detected by the model, FP represents the number of false positive instances detected as positive samples by the model, FN represents the number of false negative instances detected as negative samples by the model, P stands for precision, R stands for recall, and P(R) represents the maximum precision when the recall is r.

## 3. Experiment and Analysis

This study presents a method for detecting Chaotian peppers in complex field environments based on the YOLOv8 model. Firstly, mainstream object detection models were trained and compared, and the YOLOv8 model with the best performance was selected. Subsequently, the YOLOv8 was initially optimized by adding BiFPN, LSKNet, and FasterNet modules, and then continuously optimized by adding EMBC, EMSCP, DAttention, MSBlock, Faster different attention, and lightweight modules on the basis of initial optimization. The CIoU, Inner CioU, Inner GIoU, and inner_mpdiou loss functions of the optimized YOLOv8 model were experimentally compared, and different scale factors were adjusted to select the optimal loss function and the best scale factor to further improve the overall detection performance of the model. Finally, the optimized YOLOv8 model was used to detect Chaotian peppers in complex field environments, and the detection results were visually analyzed.

### 3.1. Basic Model Test of Chaotian Pepper Detection

To test the performance of the YOLOv8 model in Chaotian pepper recognition, the experiment used the Chaotian pepper training set and test set, divided in Section 2.1.3 of the paper. First, the original YOLOv3, YOLOv5, YOLOv6, and YOLOv8 object detection models were trained on the training set using the Adam optimizer, with a learning rate initialized to 0.001, a batch size of 8, and trained for 200 epochs. After training, the trained models were tested on the test set to calculate the accuracy, recall, mAP, and FPS of the base models, and their performance in Chaotian pepper detection was compared. The detection accuracy, recall, mAP, and FPS values of the base models are shown in Figure 13. In the figure, A represents images taken in sunny weather, B represents images taken on cloudy days, C represents stem and leaf obstruction, and D represents a mixture of green and red peppers.

Based on Figure 13, in terms of Chaotian pepper recognition, these four models perform best for Chaotian pepper recognition taken on sunny days, followed by Chaotian pepper taken on overcast days, and the worst when the Chaotian pepper stems and branches are obstructed by leaves. Among them, the YOLOv6 model only achieves a precision of 52.4% and a recall of 68.7% for the detection of Chaotian pepper with stem and branch occlusion, indicating poor performance, which may be related to the limited amount of Chaotian pepper stem and branch data or discrepancies in data annotation. Compared to the other three models, the YOLOv8 model shows comprehensive advantages in terms of accuracy and speed. The specific evaluation results of the four models are shown in Table 3.

From Table 3, it can be seen that in the comparative experiment that the precision of the YOLOv8 model can reach 77.9%; the recall can also reach 71.0%; the mAP can reach 81.6%, with its computational cost being 8.1GFLOPs, which is relatively small; and the FPS can reach 493.7. Considering the above evaluation indicators of various models, choosing YOLOv8 for subsequent experiments is advisable.

### 3.2. Parameter Optimization Experiment of Chaotian Pepper Detection Model

In this study, pepper data in the field environment were used, taken from multiple perspectives. Due to the significant scale variation of peppers, it is difficult to extract features. Therefore, different feature extraction and fusion methods were attempted in this study to experiment with peppers. The effects of different modules on the basic model were compared, and the optimal module was selected as the basis for subsequent continuous optimization. The replaced feature extraction and fusion modules include BiFPN, LSKNet, and FasterNet. The experimental results are shown in Table 4.

According to Table 4, the precision of the base model increases to 73.7% after adding the LSKNet module, which is the lowest value among the added modules. The recall value of 72.5% is in a mediocre position, and the mAP of 80.5% is also the lowest among the various modules. The increase in computational complexity to the maximum value implies the highest hardware requirements. Therefore, adding the LSKNet module has a negative effect on the optimization of the model. This is because the LSKNet module, although utilizing multi-scale feature information, has a stronger perception ability for small targets and is more suitable for remote sensing data. On the other hand, the addition of the FasterNet module enhances the detection accuracy to some extent, with the precision increasing from 77.9% to 79.4%, a 1.5 percentage point increase. Similarly, the recall value also increases from 71.0% to 72.5%, a 1.5 percentage point increase. The computational complexity increases from 8.1 GFLOPs to 10.7 GFLOPs, a 32.1% increase, indicating relatively high hardware requirements. After adding the BiFPN to the base model, the precision increases from 77.9% to 81.8%, a 2.9 percentage point increase, which is the highest among all added modules. The recall value also increases from 71.0% to 73.3%, a 2.3 percentage point increase, the largest improvement among all added modules. The mAP also increases by 1.0 percentage point, while the computational complexity decreases to some extent. Experiments show that among all added modules, the addition of the BiFPN module has the best optimization effect on the base model. Therefore, the BiFPN module is finally chosen as the optimization module for the YOLOv8 model, and subsequent continuous optimization experiments are carried out based on this.

### 3.3. Lightweight Optimization Experiment of Chaotian Pepper Detection

In this research, we used images of pepper plants in the field under complex conditions such as uneven lighting, stem obstruction, leaf obstruction, fruit mutual obstruction, and complex mixtures of green and red fruits. By introducing the BiFPN module, the problem of information loss when extracting features of different scales is partially solved, and it has different degrees of an elimination effect on lighting, color changes, and obstruction. To improve the comprehensive detection performance of the model, after adding the BiFPN module, this study continues to add different attention and lightweight modules, such as EMBC, EMSCP, DAttention, MSBlock, and FasterNet in the C2f module, and provides experimental comparisons of their different effects. The performance of various indicators of the basic model after adding different modules is shown in Figure 14.

According to Figure 14, after adding different modules in YOLOv8-BiFPN, the indicators of MSBLOCK module are lower than other modules, which may be related to the fact that this module aims to enhance the ability of real-time object detectors to handle objects of different scales. In the Chaotian pepper dataset, the scale differences are not very obvious, so its effect is not very significant in this dataset. The other four different modules have improved the precision, recall, mAP_0.5, and mAP_0.5:0.95 of the base model. From the latter half of the model (epoch 150~200), EMSCP has a slight advantage over the other modules EMBC, DAttention, and Faster, possibly because EMSCP concatenates feature maps at different scales and obtains richer features through channel fusion technology. Experiments show that using the YOLOv8-BiFPN-EMSCP model for detecting Chaotian peppers in large field environments has good results. The specific data indicators of this optimization method are shown in Table 5.

According to Table 5, the precision of using the YOLOv8-BiFPN-EMSCP model reaches 79%, which is the best among the added five modules. The mAP is 83.2, which is higher than the effect of adding other modules in the experiment. The recall indicator reaches 75.3%, slightly lower than YOLOv8-BiFPN-DAttention. The mAP at 0.5:0.95 is 57.2, which is lower by 0.4 compared to adding the EMBC module. The change in this value can be ignored in practical application. Considering the changes in various indicators, the YOLOv8-BiFPN model with the added EMSCP module achieves better comprehensive indicators. The evaluation index changes from the selection of the basic model to the determination of the final optimized model are shown in Table 6.

Table 6 shows that after adding the EMSCP module to the baseline model, the precision value of the model increased from 77.9% to 79.0%, an increase of 1.1 percentage points; the recall metric increased from 71.0% to 75.3%, an increase of 4.3 percentage points; the mAP metric increased from 81.6% to 83.2%, an increase of 1.6 percentage points; and the mAP0.5:0.95 showed the smallest relative improvement, increasing from 56.6% to 57.2%, a 0.6 percentage point increase. All evaluation metrics showed improvement to varying degrees, with improvements ranging from 0.6 to 4.3 percentage points. The experimental results indicate that the addition of the EMSCP module has a positive effect on the performance improvement of the baseline model.

### 3.4. Optimization Model Ablation Experiment of Chaotian Pepper Detection

After module optimization, attention mechanism, and lightweight module optimization, the comprehensive performance of YOLOv8 in this study improved. In order to determine the optimization effect of each module on the base model, this section conducts ablation experiments on the pepper detection optimization model. In this study, BiFPN, EMSCP, and BiFPN-EMSCP modules are added, respectively, and the optimization effects of each module are determined through experiments.

The BiFPN module introduces bidirectional flow of feature information, effectively solving the problem of information loss in extracting features of different scales in traditional feature pyramid networks. BiFPN introduces learnable weights and fuses the upsampled and downsampled feature maps layer by layer, introducing lateral and vertical connections, so that features of different scales can be better fused and utilized. The EMSCP module, on the other hand, is a multi-scale convolution network architecture that captures different features in images by applying multi-scale convolution operations on input images at different scales, concatenating feature maps at different scales to obtain richer feature representations, thus helping to improve the model’s perceptual and expressive capabilities, which is especially suitable for processing images with features of different scales. This study compares the experimental results of these two models in three experiments. The results are shown in Figure 15.

From Figure 15, it can be seen that after adding various modules and combinations in YOLOv8, the performance indicators of various experimental schemes are relatively close. The precision value stabilizes around 80% after epoch 50, with some oscillations. The experiment performance with the addition of the BiFPN module is slightly higher than that of other schemes, and by epoch 200, the value stabilizes around 80%. On the other hand, the recall indicator shows significant oscillations in each scheme; however, with the addition of BiFPN and EMSCP, the changes are flatter, with values around 70%. The mAP_0.5 indicator remains relatively stable, with values around 80% after epoch 50. From epoch 175 to 200, the schemes with the addition of BiFPN and BiFPN-EMSCP consistently oscillate around 80%, higher than other schemes. The mAP_0.5:0.95 indicator follows a similar trend to mAP_0.5, with slightly higher oscillations, and stabilizes around 55%. The specific numerical changes of each indicator in each experimental scheme are shown in Table 7.

From Table 7, it can be seen that among the three proposed schemes, the precision of the YOLOv8-BiFPN-EMSCP scheme reaches 79.0%, only 2.8 percentage points lower than the highest value. Additionally, its recall value is 75.3%, which is 4.3 percentage points higher than YOLOv8. The mAP_0.5 and mAP_0.5:0.95 metrics are the highest among the three schemes, with values of 83.2 and 57.2, respectively. The model’s GFLOPs is 7.0 and the model size is 4.0 M, both of which are the best among the three schemes. The FPS value is 301.4. Although the metrics are slightly lower than the baseline model, they still meet the real-time requirements for applications. Taking into account the changes in various indicators of the YOLOv8-BiFPN-EMSCP scheme, the experimental scheme with BiFPN-EMSCP added has the best optimization effect on the baseline model.

### 3.5. Chaotian Pepper Detection Model Parameter Optimization Experiment

The IoU loss function has wide applications in computer vision tasks. The relationship between IoU variation and bounding box size variation has a certain impact on the detection efficiency of models. There is only a scale difference between the auxiliary box and the actual box. In the regression process, the trend of IoU value change of the auxiliary box is consistent with the trend of IoU value change of the actual box, which can reflect the quality of the regression result of the actual box. In this study, CIoU, Inner CioU, Inner GIoU, and inner_mpdiou were selected to adjust different scale factors for comparative experiments in order to select the optimal loss function and the best scale factor so that the model can converge better. The calculation formula of the IoU loss function is as follows:(13)$$IoU^{inner}=\frac{inter}{union}$$

The parameters in Formula (13) are calculated according to Formulas (14) to (15).
(14)$$union=\left(w^{gt}\times h^{gt}\right)\times {\left(ratio\right)}^{2}+\left(w\times h\right)\times {\left(ratio\right)}^{2}-inter$$
(15)$$inter=\left(\min\left(b_{r}^{gt},b_{r}\right)-\max\left(b_{l}^{gt},b_{l}\right)\right)\cdot \left(\min\left(b_{b}^{gt},b_{b}\right)-\max\left(b_{t}^{gt},b_{t}\right)\right)$$

The parameters in Formula (14) and Formula (15) are calculated according to Formula (16) to Formula (19).
(16)$$b_{t}=y_{c}-\frac{h\times ratio}{2},b_{b}=y_{c}+\frac{h\times ratio}{2}$$
(17)$$b_{l}=x_{c}-\frac{w\times ratio}{2},b_{r}=x_{c}+\frac{w\times ratio}{2}$$
(18)$$b_{t}^{gt}=y_{c}^{gt}-\frac{h^{gt}\times ratio}{2},b_{b}^{gt}=y_{c}^{gt}+\frac{h^{gt}\times ratio}{2}$$
(19)$$b_{l}^{gt}=x_{c}^{gt}-\frac{w^{gt}\times ratio}{2},b_{r}^{gt}=x_{c}^{gt}+\frac{w^{gt}\times ratio}{2}$$

In the formulas below, w and h represent the width and height of the anchor box, $\left(x_{c},y_{c}\right)$ represent the center coordinates of the anchor box; $w^{gt}$ and $h^{gt}$ represent the width and height of the target box; $\left(x_{c}^{gt},y_{c}^{gt}\right)$ represent the center coordinates of the target box; and ratio is the scale factor, typically in the range of [0.5, 1.5]. The calculation formulas for CIoU, Inner CioU, Inner GIoU, and inner_mpdiou loss functions are as follows:(20)$$L_{Inner}-CIoU=L_{CIoU}+IoU-IoU^{inner}$$
(21)$$L_{Inner}-GIoU=L_{GIoU}+IoU-IoU^{inner}$$

In the experiment, Inner CIoU was first selected with a ratio factor set to 0.8. The model’s evaluation metrics were 76.7, 76.3, and 82.5. After adjusting the ratio factor to 1.0, the corresponding metrics became 81.4, 71.5, and 81.2, showing both improvements and declines. The accuracy increased by 4.7 percentage points, while the recall rate and mAP slightly decreased. Then, the Inner GioU loss function was selected with ratio factors of 0.6, 0.7, 0.8, 0.85, and 0.9 for comparative experiments. The precision showed an increasing trend ranging from 79.1 to 81.2, while the recall rate fluctuated between 68.4 and 78.1 and the mAP varied between 82 and 83.2. The mAP0.5:0.95 changed from 55.9 to 57. In the comparative experiment of selecting inner_mpdiou, the ratio factor was adjusted to 0.7, and the corresponding metrics were 0.763, 0.773, 0.819, and 56.4. Considering all factors, the CIoU loss function was ultimately chosen to help the model converge better. Detailed information of each loss function and ratio factor adjustment are shown in Table 8.

According to Table 8, each loss function shows different changes in various indicators, making it difficult to make a direct judgment on overall performance. In order to better demonstrate the performance of each loss function, this experiment tested their performance on both the training set and validation set, with the results shown in Figure 16.

From Figure 16, it can be seen that the IndianRed curve shows lower overall performance than the other eight lines. The decrease rate is relatively fast in the first 25 rounds of training, with the loss function value quickly dropping from around 3.6% to around 2.2%. The rate of performance loss reduction from 25 rounds to 200 rounds decreases but overall performance continues to improve. By the 185th round, there is a sudden change in the rate of decrease, with the value decreasing rapidly by around 0.2 percentage points. After that, the efficiency of the loss function gradually stabilizes, with the red curve’s value stabilizing at around 1.2%. By adjusting the scale factor of this loss function between 0.6, 0.7, 0.8, 0.85, and 0.9, the loss function curves IndianRed, Thistle, DarkGray, DarkKhaki, and Plum are formed. The overall performance trends of these curves are the same as the red curve. To select the optimal loss function, inner Ciou and inner_mpdiou functions are chosen separately, with different scale factors set to form other curves in the graph. Among them, the DarkGray curve represents the performance of inner_mpdiou at 0.7, which is better than the IndianRed curve and Thistle curve overall, but inferior to the performance of other loss functions in this experiment. The curve representing inner Ciou has the best overall performance, with the SteelBlue curve at the bottom of this experiment, indicating that all its indicators are optimal. To further verify the effectiveness of the choice of loss function, the performance of the corresponding loss functions on the validation set was tested, as shown in Figure 17.

From Figure 17, it can be seen that the performance of each loss function on the validation set is consistent with that on the test set, with slightly larger fluctuations. The optimal loss function also performs the best on the test set, so this loss function is selected as the experimental parameter for optimizing the final model.

### 3.6. Visualization Analysis of Detection Results of the Final Optimized Model on the Test Set

To intuitively demonstrate the areas of focus by the model and the role of attention mechanisms in guiding the network to key areas, this study utilized Grad-CAM [38] for visualization to present the decision-making process from the second layer to the final layer of the model. Grad-CAM generated the heatmaps for YOLOv8 and YOLOv8 with an added optimization mechanism, as shown in Figure 18.

In the heatmap, the intensity of colors represents the model’s confidence in detecting targets, with the red areas highlighting the key regions where the model predicts the target’s position. As shown in the figure, both models are capable of accurately distinguishing between targets and backgrounds. In Figure 18(a1,a2), the model’s focus is on the Chaotian pepper region of the image, indicating that the model has successfully learned the most distinctive features of the Chaotian pepper, which helps improve the accuracy of Chaotian pepper detection and reduce false detections. Additionally, after adding the Optimization mechanism, the colors in the heatmap become darker, indicating higher confidence. In the heatmap of group b, it can be seen from panel c that YOLOv8 does not pay enough attention to the Chaotian pepper in the lower-left corner, which can easily lead to missed detections. However, after adding the optimization mechanism, the attention to this is significantly enhanced. The heatmap of group d shows that although YOLOv8’s attention points can distinguish targets from the background, more targets will still be overlooked. With the addition of the optimization mechanism, the network can more accurately focus the attention points on the target Chaotian pepper.

The results of this experiment show that the introduction of optimization mechanism makes the network’s focus on the critical region more explicit and further improves the model’s recognition accuracy of the target. This has a positive impact on optimizing the Chaotian pepper detection task.

## 4. Discussion

### 4.1. Impact Analysis of Dataset Richness on Object Detection Performance

The improved YOLOv8 object detection model proposed in this study achieves real-time and efficient recognition of Chaotian peppers in complex field environments, where the precision value of the model increased from 77.9% to 79.0%, an increase of 1.1 percentage points; the recall metric increased from 71.0% to 75.3%, an increase of 4.3 percentage points; and the mAP metric increased from 81.6% to 83.2%. The detection speed can reach 339 FPS; although it is slightly lower compared to the baseline model, it still meets the accuracy and speed requirements for practical picking. However, due to the uneven growth stages of bell peppers in natural environments, the Chaotian peppers in the field usually have different appearance characteristics and morphological structures, which may affect the model’s detection accuracy. Therefore, future research could consider adding data of Chaotian peppers in different growth stages to train the model, while paying more attention to the quality of training data to ensure accuracy and diversity, enhance the neural network’s generalization ability, and improve its capability to adapt to Chaotian peppers in different growth stages.

### 4.2. Impact Analysis of Different Backbone Networks on the Target Detection Performance of Chaotian Pepper

In this study, various optimization methods were used to improve the detection performance of the base model. The precision, recall, and mAP metrics increased by 1.1, 4.3, and 1.6 percentage points, respectively. In practical production, in addition to pursuing metrics such as precision and recall, there are also requirements for the model’s size, running speed, and other aspects based on different deployment needs. Therefore, various optimization methods can be adopted according to actual production requirements. The application of lightweight networks is very common, so the optimization methods in this study can be combined to continue lightweight network optimization research. Ghost, Transformer, and MobileNetv3 are currently the mainstream lightweight networks. Among them, Transformer and Ghost networks have high accuracy but are slower than MobileNetv3, which has faster speed but lower accuracy. Therefore, the appropriate network should be selected for optimization based on specific requirements such as accuracy, speed, and resource constraints in different scenarios. Adjusting the backbone network is a common optimization method. Following the optimization method of the backbone network in YOLOv5, a new FS-MobileNetV3 network proposed by Huang Lei et al. [38] replaces the CSPDarknet backbone network in the original network to extract feature images. Compared with the original model, the improved model only reduces the average accuracy by 0.37 percentage points and increases the detection speed by 11 FPS, meeting the requirements for deployment on mobile devices in different scenarios. Shao Xiaoqiang et al. [39] use the improved lightweight network ShuffleNetV2 to replace the original CSP-Darknet53 backbone network in YOLOv5s and integrate the Transformer self-attention module into the improved ShuffleNetV2. The improved detection model has an average detection accuracy increase of 5.2 percentage points and a speed increase of 21 percentage points compared to the original YOLOv5s model. Based on these research results, future research can be conducted as follows: replace the backbone network of the YOLOv8 model with a lighter Ghost, Transformer [40], or MobileNetv3 [41] backbone network, and train and test them with the same training parameters as before to meet different production needs.

## 5. Conclusions

This study aims to achieve efficient detection of Chaotian pepper in the field under complex environments, using four mainstream object detection algorithms for training comparison and selecting YOLOv8 with the best detection performance for experimentation. Based on this, preliminary optimizations of YOLOv8 were conducted by adding BiFPN, LSKNet, and FasterNet modules, followed by continuous optimizations with EMBC, EMSCP, DAttention, MSBlock, Faster different attention, and lightweight modules. The CIoU, Inner CioU, Inner GIoU, and inner_mpdiou loss functions of the YOLOv8 optimized model were experimentally compared by adjusting different scaling factors to select the optimal loss function and the best scaling factor in order to further improve the overall detection performance of the model and achieve high-precision real-time detection of pepper in complex field environments. The conclusions are as follows:(1)Compared to YOLOv3, YOLOv5, and YOLOv6 models, the precision, recall, and mAP metrics of the YOLOv8 model can reach 77.9%, 71.0%, 81.6%, and the inference speed can reach 493.7 FPS, with higher detection accuracy and inference speed.(2)Adding the BiFPN module to the YOLOv8 model outperforms the addition of the LSKNet and FasterNet modules. We incorporate different attention and lightweight modules like EMBC, EMSCP, DAttention, MSBlock, and Faster, where adding the EMSCP module yields the best results, with the model’s precision, recall, and mAP metrics improving by 1.1%, 4.3%, and 1.6%, respectively. Comparative experiments on the loss functions CIoU, Inner CioU, Inner GIoU, and inner_mpdiou confirm that CIoU has better convergence effects, with the optimized model running at an inference speed of 393 FPS—although it has decreased, it still meets real-time requirements.(3)Under the influence of factors such as stem obstruction, leaf and fruit obstruction, different scales, and sunny and cloudy weather, the optimized YOLOv8 model can still efficiently detect Chaotian peppers. The precision, recall, and mAP indicators for Chaotian pepper detection reach 79.0%, 75.3%, and 83.2%, respectively. The GFLOPs are reduced by 13.6%, the model size is reduced to 66.7% of the base model, and the FPS value of the optimized model reaches 301.4. This method has good stability and can meet the requirements of on-site Chaotian pepper picking operations, providing a technical reference for the detection of crops in similar environments and the realization of intelligent picking of Chaotian peppers.

## Figures and Tables

**Figure 1 sensors-24-05632-f001:**
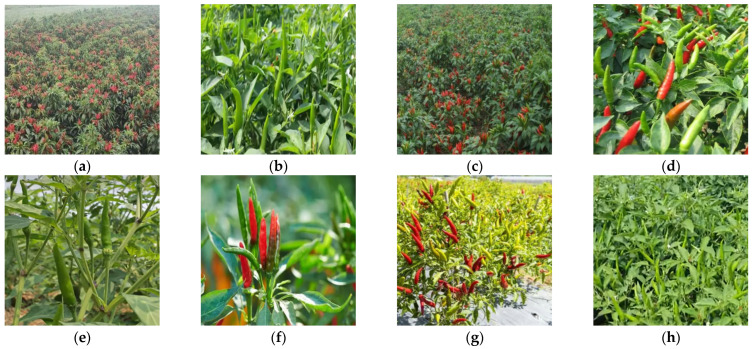
Images of daylilies collected in the complex field environment. (**a**) Real scene base; (**b**) normal weather; (**c**) sunny backlight; (**d**) green and red mixed; (**e**) stem and leaf obscuring; (**f**) front view angle; (**g**) top view angle; (**h**) side view angle.

**Figure 2 sensors-24-05632-f002:**
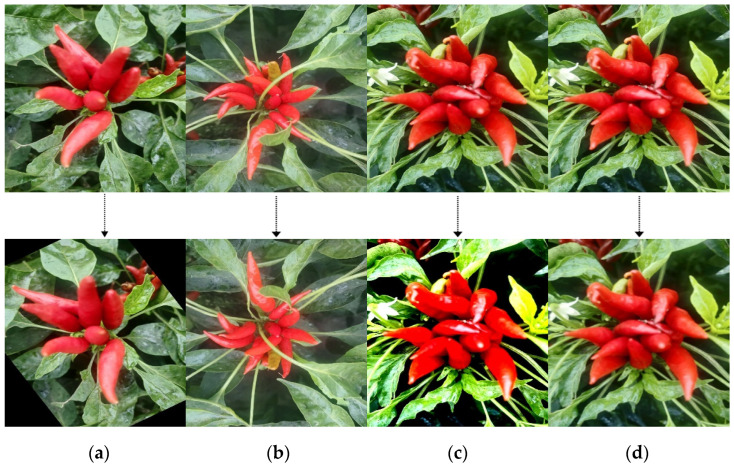
Data augmentation on Chaotian pepper images. (**a**) Random rotation; (**b**) vertical flip; (**c**) contrast enhancement; (**d**) Gaussian blur.

**Figure 3 sensors-24-05632-f003:**
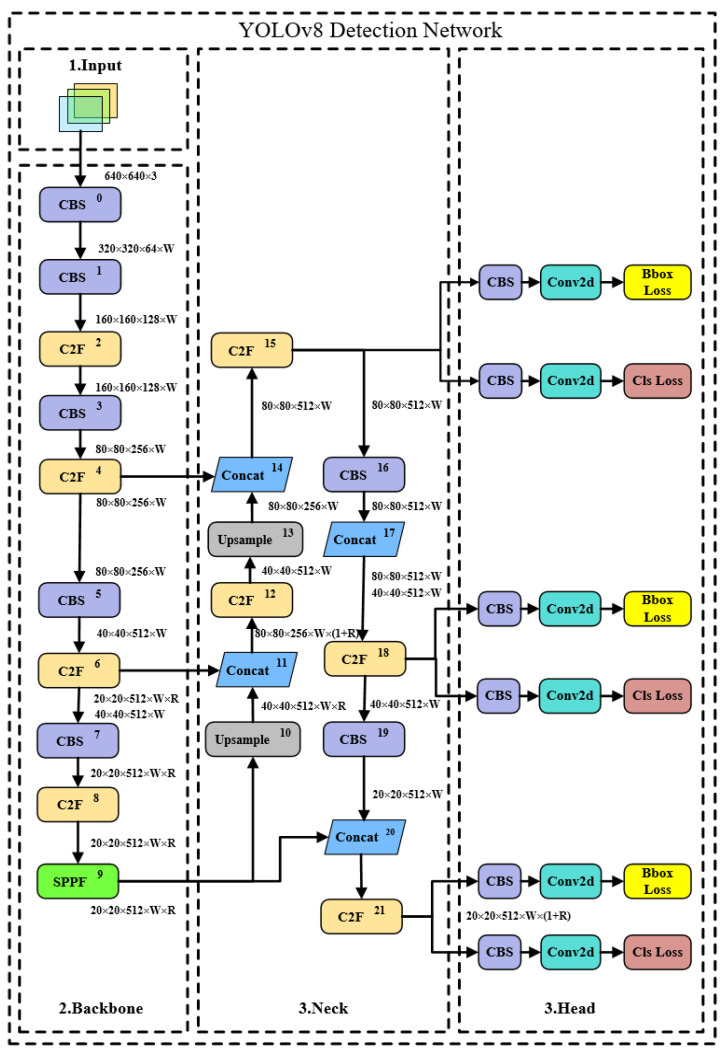
YOLOv8 network framework.

**Figure 4 sensors-24-05632-f004:**
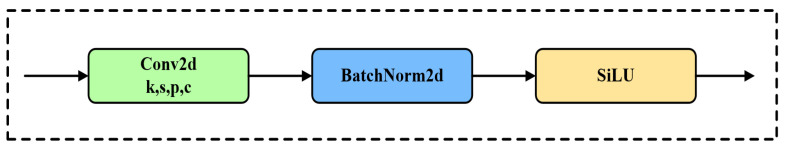
CBS Convolutional Block. k represents the kernel size, s represents the stride, p represents the padding size, and c represents the input/output channels.

**Figure 5 sensors-24-05632-f005:**
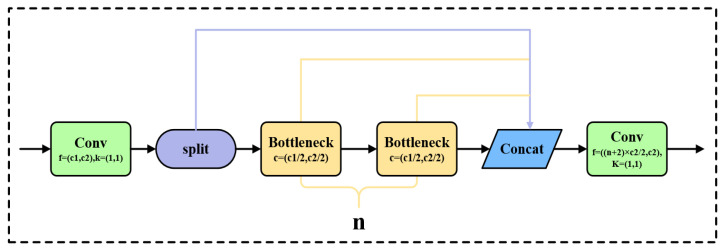
C2f Convolutional Block.

**Figure 6 sensors-24-05632-f006:**
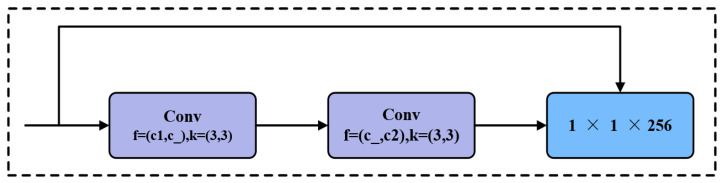
Structure of Bottleneck module.

**Figure 7 sensors-24-05632-f007:**
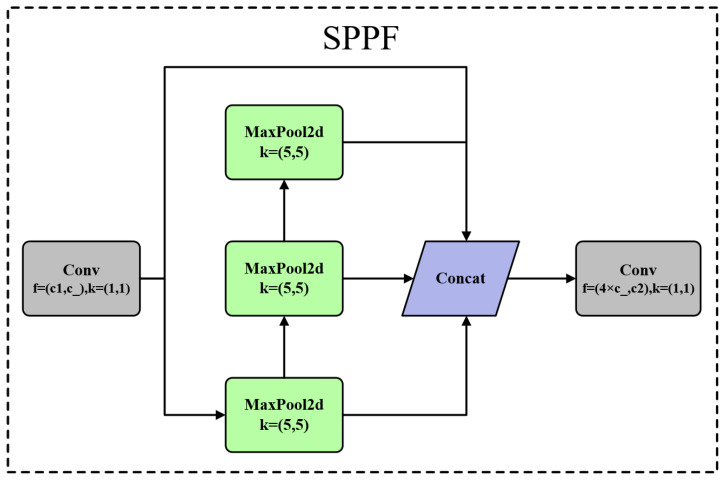
SPPF structure diagram.

**Figure 8 sensors-24-05632-f008:**
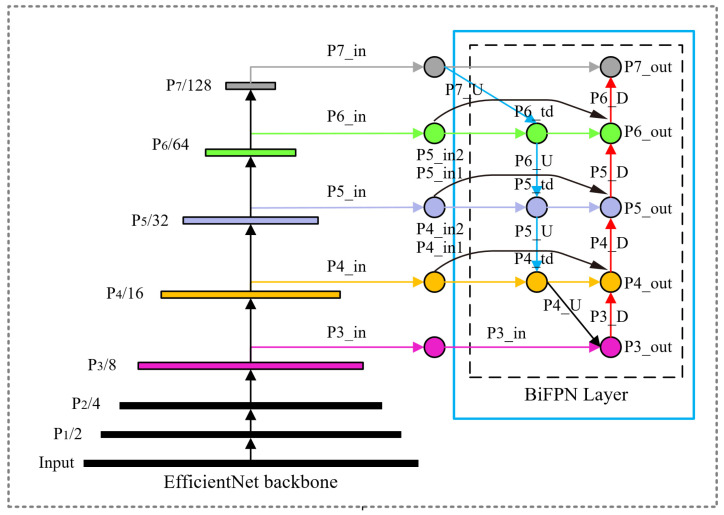
BiFPN structure diagram.

**Figure 9 sensors-24-05632-f009:**
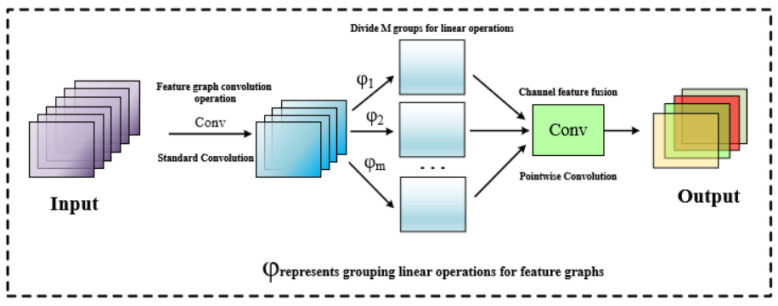
EMSCP structure diagram.

**Figure 10 sensors-24-05632-f010:**
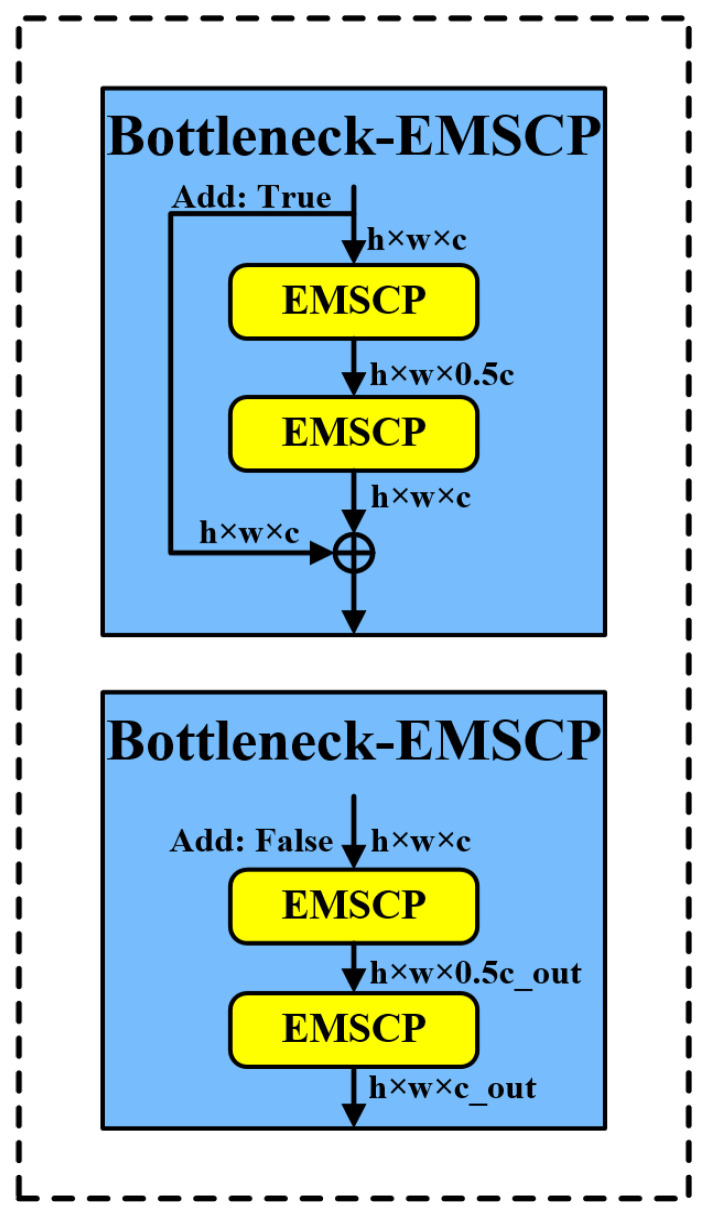
Bottleneck-EMSCP structure diagram.

**Figure 11 sensors-24-05632-f011:**
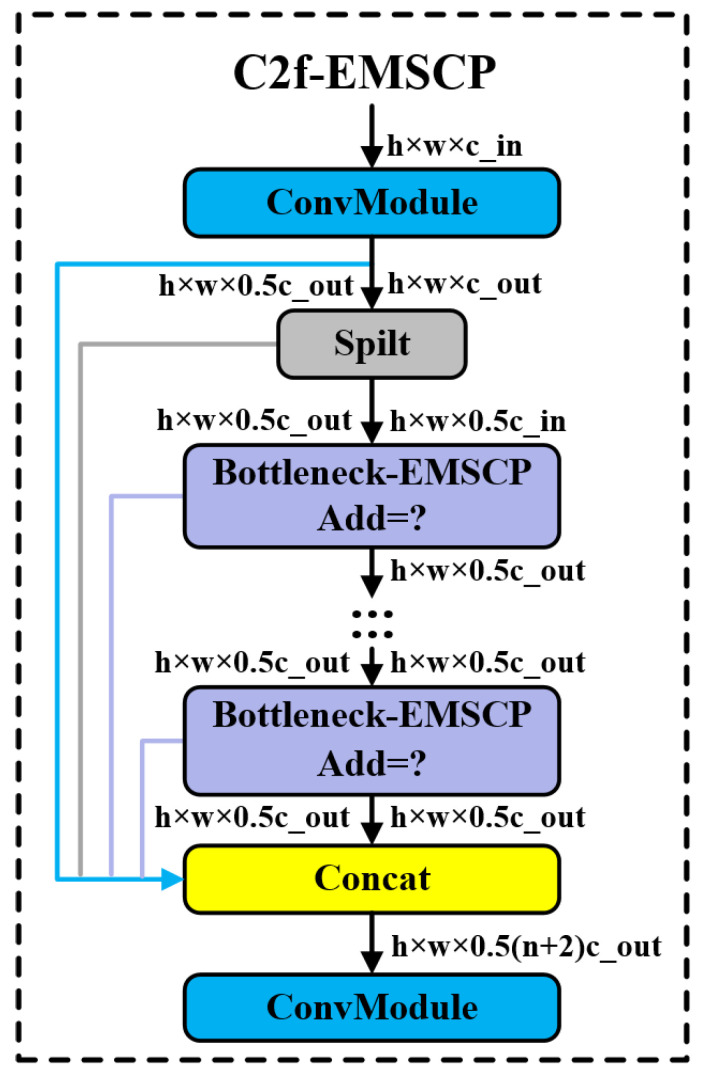
C2f-EMSCP structure diagram.

**Figure 12 sensors-24-05632-f012:**
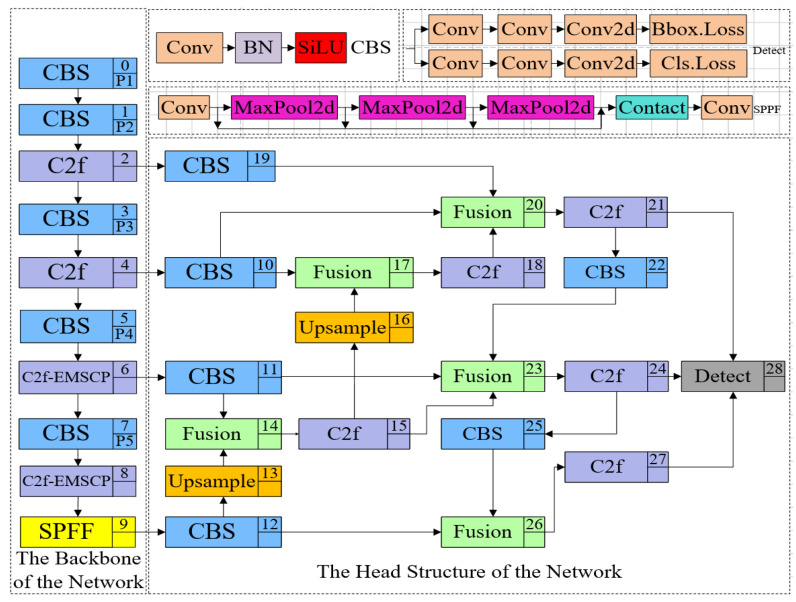
Overall structure diagram of the optimized model.

**Figure 13 sensors-24-05632-f013:**
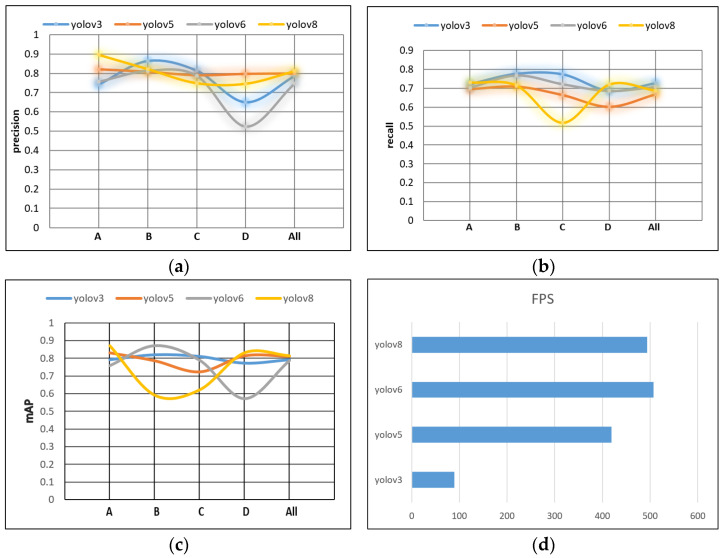
Comparison of detection results of YOLOv3, YOLOv5, YOLOv6, and YOLOv8 models. (**a**) Precision; (**b**) recall; (**c**) mAP; (**d**) FPS.

**Figure 14 sensors-24-05632-f014:**
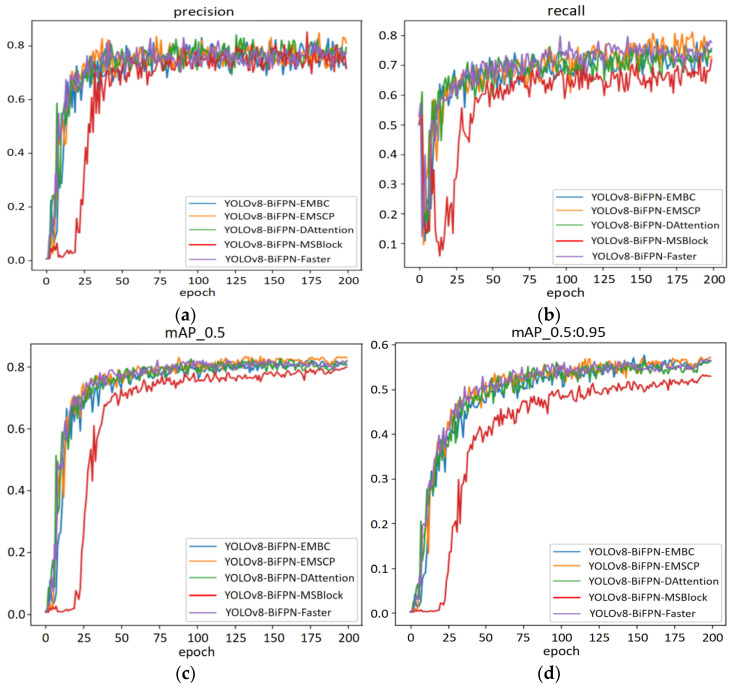
Performance comparison of YOLOv8 model with different modules. (**a**) Precision; (**b**) recall; (**c**) mAP_0.5; (**d**) mAP_0.5:0.95.

**Figure 15 sensors-24-05632-f015:**
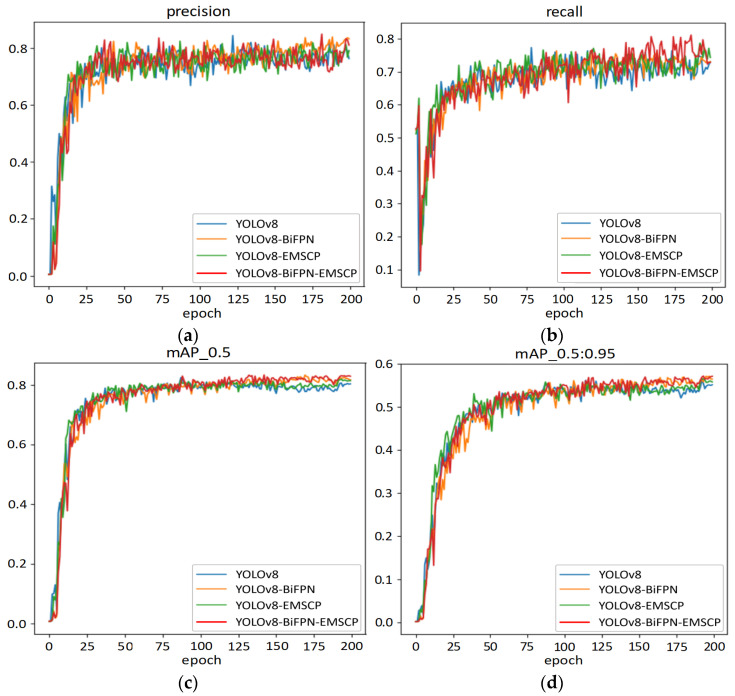
Performance comparison of YOLOv8 model ablation experiments. (**a**) Precision; (**b**) recall; (**c**) mAP; (**d**) mAP0.5:0.95.

**Figure 16 sensors-24-05632-f016:**
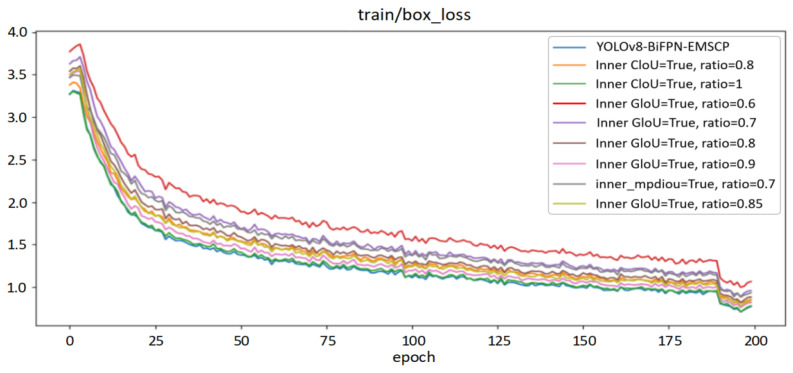
Performance comparison of different loss functions of YOLOv8 model on the training set.

**Figure 17 sensors-24-05632-f017:**
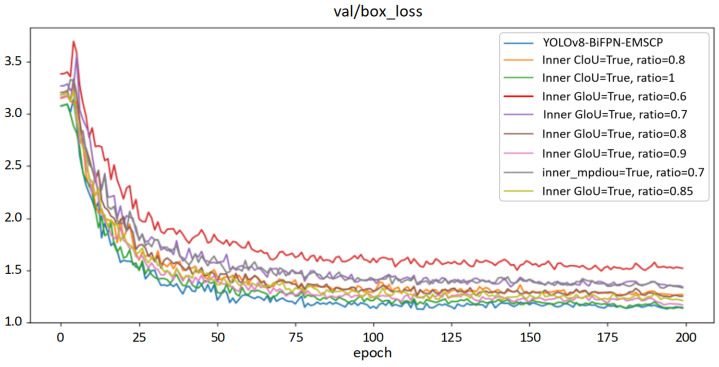
Performance comparison of different loss functions of YOLOv8 model on the validation set.

**Figure 18 sensors-24-05632-f018:**
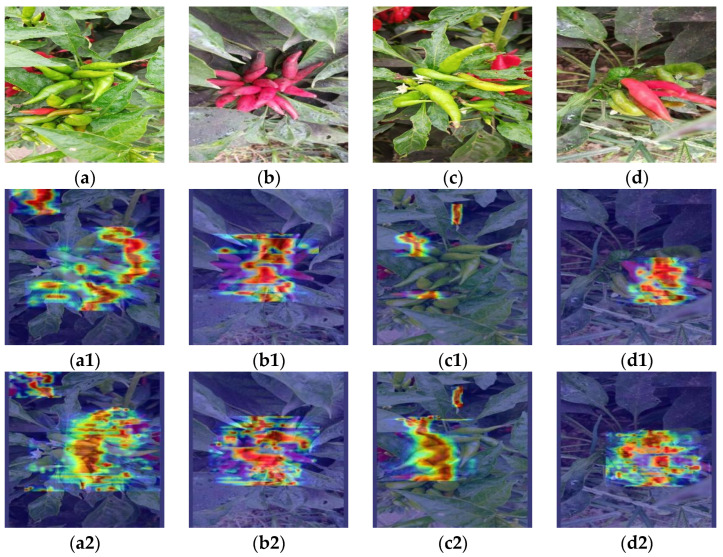
Partial visualization display of YOLOv8 optimized model detection results. (**a**–**d**) represents the image to be processed; (**a1**–**d1**) represents the heatmap of the model without the optimization mechanism; (**a2**–**d2**) represents the heatmap of the model with the optimization mechanism.

**Table 1 sensors-24-05632-t001:** Distribution table of four scenes of labels in the pepper dataset.

Scene	Training Set Labels	Validation Set Labels	Test Set Labels	Training Set	Validation Set	Test Set
Sunny Scene	15,778	4024	2252	1834	510	204
Obstructed Scene	4450	1124	635	801	252	135
Mixed Scene	4124	1140	588	785	218	133
Cloudy Scene	2700	680	384	122	51	30
Total	27,052	6968	3859	2800	800	400

**Table 2 sensors-24-05632-t002:** Model training parameters.

Parameters	Value
Input image resolution	640 × 640
Iterations	200
Batch size	8
Initial learning rate	0.001
Learning rate momentum	0.937
Weight decay coefficient	5 × 10^−4^

**Table 3 sensors-24-05632-t003:** Results of YOLOv3, YOLOv5, YOLOv6, and YOLOv8 model tests.

Model	Classes	Precision (%)	Recall (%)	mAP (%)	GFLOPs	FPS
YOLOv3	A	0.741	0.723	0.792	282.2	178
	B	0.864	0.778	0.819		
	C	0.814	0.775	0.810		
	D	0.649	0.687	0.773		
	All	0.786	0.728	0.792		
YOLOv5	A	0.820	0.696	0.830	7.1	48
	B	0.808	0.710	0.785		
	C	0.790	0.665	0.722		
	D	0.797	0.603	0.813		
	All	0.799	0.669	0.810		
YOLOv6	A	0.756	0.704	0.758	11.8	33
	B	0.813	0.768	0.872		
	C	0.788	0.722	0.789		
	D	0.524	0.687	0.570		
	All	0.747	0.708	0.787		
YOLOv8	A	0.897	0.731	0.870	8.1	14
	B	0.822	0.715	0.590		
	C	0.748	0.518	0.620		
	D	0.745	0.720	0.830		
	All	0.779	0.710	0.816		

A: Taken on a sunny day. B: Taken on a cloudy day. C: Stems and leaves obscuring. D: Mixture of green and red.

**Table 4 sensors-24-05632-t004:** Experimental results of YOLOv8 model with different network replacements.

Model	Precision (%)	Recall (%)	mAP (%)	mAP 0.5:0.95 (%)	GFLOPs
YOLOv8	77.9	71.0	81.6	56.6	8.1
YOLOv8-BiFPN	81.8	73.3	82.6	57.1	7.1
YOLOv8-LSKNet	73.7	72.5	80.5	55.0	19.7
YOLOv8-fasternet	79.4	72.5	81.9	55.2	10.7

**Table 5 sensors-24-05632-t005:** Experimental results of adding different attention mechanisms to the YOLOv8 model.

Model	Precision (%)	Recall (%)	mAP (%)	mAP0.5:0.95
YOLOv8-BiFPN-EMBC	75.2	74.2	82.0	57.6
YOLOv8-BiFPN-EMSCP	79.0	75.3	83.2	57.2
YOLOv8-BiFPN-DAttention	75.0	76.2	82.8	56.4
YOLOv8-BiFPN-MSBlock	74.0	68.1	79.5	53.2
YOLOv8-BiFPN-Faster	78.0	74.8	81.6	57.3

**Table 6 sensors-24-05632-t006:** Comparison of the final optimization results of YOLOv8 model.

Model	Precision (%)	Recall (%)	mAP (%)	mAP0.5:0.95
YOLOv8-BiFPN-EMSCP	79.0	75.3	83.2	57.2
YOLOv8	77.9	71.0	81.6	56.6

**Table 7 sensors-24-05632-t007:** Experiment results of YOLOv8 model with BiFPN and EMSCP added.

Model	Precision (%)	Recall (%)	mAP (%)	mAP0.5:0.95	GFLOPs	Model Size	FPS
YOLOv8	77.9	71.0	81.6	56.6	8.1	6.0	446.1
YOLOv8-BiFPN	81.8	73.3	82.6	57.1	7.1	4.1	351.7
YOLOv8-LSKNet	77.6	77.3	82.0	56.6	7.9	5.8	308.1
YOLOv8- BiFPN-EMSCP	79.0	75.3	83.2	57.2	7.0	4.0	301.4

**Table 8 sensors-24-05632-t008:** Loss function and scale factor adjustment table.

Model	Precision (%)	Recall (%)	mAP (%)	mAP0.5:0.95
YOLOv8-BiFPN-EMSCP-CIou	79.0	75.3	83.2	57.2
Inner CIoU = True, ratio = 0.8	76.7	76.3	82.5	57.5
Inner CIoU = True, ratio = 1	81.4	71.5	81.2	56.6
Inner GIoU = True, ratio = 0.6	79.3	71.8	82.0	56.0
Inner GIoU = True, ratio = 0.7	79.1	78.1	82.7	57.0
Inner GIoU = True, ratio = 0.8	80.2	73.8	83.2	55.9
Inner GIoU = True, ratio = 0.9	73.3	75.5	81.9	57.0
Inner GIoU = True, ratio = 0.85	81.2	68.4	82.0	57.0
inner_mpdiou, ratio = 0.7	76.3	77.3	81.9	56.4

## Data Availability

Data are contained within the article.

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
