# Peer review of "Research on Detection Method of Chaotian Pepper in Complex Field Environments Based on YOLOv8"

_sensors, 2024, doi:10.3390/s24175632_

Round 1

Reviewer 1 Report

Comments and Suggestions for Authors

To address the need for intelligent harvesting of chili peppers, a target recognition model based on the improved YOLOv8 is proposed. The model mainly improves the Backbone and Neck network by introducing EMBC and BiFPN, and comparisons with other similar modules have been made, demonstrating certain practical value. However, the content organization and language expression of the paper need further optimization.

BiFPN should be an improvement on the original neck, but the principles of module integration are not clearly explained in the manuscript.

In the EMSCP module principle (Figure 9), the characters do not correspond to the text, and all annotations and symbols in the diagram should have corresponding explanations in the text. Similar issues exist in other diagrams as well.

Is it necessary to compare YOLOv8 with v3, v5, and v6 models since YOLOv8 is the latest model?

From Table 4, it can be seen that although the accuracy of the improved model has increased, the performance in terms of computational load and efficiency has decreased. Is this acceptable? Additionally, can GFLOPs be used to measure the model's computational load, or is FLOPs more appropriate?

In Table 7, do the FLOP and FPS values correspond to the data below?

Reviewer 2 Report

Comments and Suggestions for Authors

The review for “Research on Detection Method of Chaotian pepper in Complex Field Environments Based on YOLOv8” (authors: Yichu Duan, Jianing Li, and Chi Zou)

The article is devoted to Chaotian pepper intelligent detection by a model based on YOLOv8. For optimization, authors added BiFPN, LSKNet, and FasterNet modules. EMBC, EMSCP, DAttention, MSBlock, Faster different attention and lightweight modules were continuously optimized on the basis of initial optimization. The authors show that with module optimization, adding different attention mechanisms and lightweight modules, and comparing different loss functions and scale factors, the precision, recall rate, and mAP have increased by 1.1, 4.3, and 1.6 percentage points, GFLOPs have been reduced by 13.6%, the model size has been reduced to 66.7% of the base model, and the optimized model's FPS value reaches 301.4. This can achieve accurate and fast detection of Chaotian pepper in complex field environments, providing data and experimental references for the development of intelligent harvesting equipment for Chaotian pepper.

Major comments

Remark 1. The abstract of the article is redundant. It is necessary to reduce the volume and briefly highlight the key elements made by the authors.

Remark 2. The introduction does not reflect the completeness of the research in this area. In the article [https://doi.org/10.6919/ICJE.202405_10(5).0038] a similar comparison was already made and the result shows that the improved YOLOv5 is more effective than YOLOv8. The result exceeds the value obtained by the authors of the article using mAP metric (0.817 versus 0.816). The question arises: what new things do the authors of the article propose and how does it differ from what has already been done? Authors must clearly highlight the novelty of the work and how their result is better than what has already been done.

Technical comments

Remark 1. Lines 217, 232, 331, 449, 503, 563, 584. Figures 5, 7, 12, 14, 15, 17. The text in the figures is very small and unreadable in print. The authors need to increase the font size and check the text in the printed version of the article.

Remark 2. The authors do not have a unified style for the design of the figures and captions. It is necessary to bring everything to a unified style.

Comments on the Quality of English Language

The article contains punctuation errors and errors in the use of tense in sentences.

Round 2

Reviewer 2 Report

Comments and Suggestions for Authors

Technical drawbacks

Remark 1. Figure 13. All figures must be designed in the same style. Authors must bring the figure into line with the article's uniform style.
